# Effects of Calcination Conditions on the Formation and Hydration Performance of High-Alite White Portland Cement Clinker

**DOI:** 10.3390/ma13030494

**Published:** 2020-01-21

**Authors:** Lei Huang, Geling Cheng, Shaowen Huang

**Affiliations:** School of Materials Science and Engineering, Nanchang University, Nanchang 330031, Jiangxi Province, China; 40132911807@email.ncu.edu.cn (L.H.); 411314019012@email.ncu.edu.cn (G.C.)

**Keywords:** white Portland cement clinker, sintering temperature, sintering time, rietveld quantitative analysis, hydration performance

## Abstract

The purpose of this study was to evaluate the effects of sintering temperature and sintering time on mineral composition of high-alite white Portland cement clinker and hydration activity of the clinker. Effects of sintering temperature and sintering time on clinker mineral composition, C_3_S polymorph and size and hydration heat release rate were analyzed by X-ray diffraction (XRD), scanning electron microscope (SEM), differential scanning calorimetry&Thermogravimetric Analysis (DSC-TG) and isothermal heat-conduction calorimetry. Results shown that, with the increase of sintering temperature (1450–1525 °C) and sintering time (60–240 min), free lime (f-CaO) in clinker decreased, C_3_S grain size increased, and C_3_S crystal type changed from T3 to M type and R type, which exhibits higher symmetry. The hydration activity of different C_3_S crystals ranged from high to low as follows: T3→M1→M3→R@.

## 1. Introduction

Cement-based materials have a lot of advantages such as performance stability, safety and durability, construction convenience and are in general considered environmentally friendly. White Portland cement is the main raw material of cement-based materials, which have only three main phases: C_3_S, C_2_S, and C_3_A, as compared to ordinary Portland cement.

C_3_S is the main component of clinker, and its hydration product is the main source of cement’s strength, which determines the key performance of cement clinker. Developments in understanding of C_3_S crystal structure are explained below:

In the 1950s, Jeffery studied the crystal structure of pure C_3_S and Al, Mg doped alite, whose crystal types were T1 and M3, respectively. To accomplish this, he derived for the first time nine pseudohexagonal cells [1]. In 1975, Golovastikov derived a representative T1 crystal structure consisting of three different groups of hexagonal planes, each containing a class of Ca polyhedral [2]. In 2003, De Noirfontaine proposed a M1 model, where he pointed out the modulation vector in the M3 superstructure based on the <M> subcrystal cell, and summarized the relationship among several crystal superstructures T1, M1 and M3 [3].

Various crystal types of C_3_S can be distinguished by the diffraction peaks in 2 very narrow angle ranges of XRD (2theta = 32~33° and 2theta = 50~52°) [4,5,6,7]. According to the existing literature, there are 7 main C_3_S crystal phases depending on the temperature and impurities: three triclinic phases (T1, T2 and T3), three monoclinal phases (M1, M2, M3) [8,9,10] and one rhombic phase(R). Below their transformation relationship [11,12,13,14,15]:
T1⇔T2⇔T3⇔M1⇔M2⇔M3⇔R.

Compared to ordinary Portland cement, high-alite Portland cement is designed with high-calcium, high-silicon, contents and almost does not contain any iron. For that reason, liquid phase appearance temperature, liquid volume, viscosity and other properties are much more different from that of the ordinary Portland cement. All these differences are due to the fact that there is no C_4_AF in the process of clinker formation in white Portland cement. Therefore, the infiltration degree of liquid phase to solid phase particles, solubility of solid phase in liquid phase, and the filling degree of the liquid phase to the solid phase are completely different, and hence have great effects on C_3_S’s formation and hydration activity [16,17]. Consequently, it is of great significance to explore sintering temperature and sintering time for the transformation rule as well as crystal characteristics and hydration activity of C_3_S in white Portland cement clinker, as outcoming results can be used to guide industrial production.

## 2. Materials and Methods

### 2.1. Sample Preparation

In order to eliminate the interference of impurity ions, pure CaCO_3_, SiO_2_ and Al_2_O_3_ were used to prepare raw materials. The composition of raw material and mineral composition (according to Bogue [18]) of cement clinkers are given in Table 1.

Raw material was molded with 8% by weight of deionized water and applying 10 MPa pressure in a Φ30 mm mold. In order to study the effects of sintering temperature on a cement clinker, prepared samples were firstly preheated for 30 min in a 500 °C electric muffle furnace, then rapidly transferred to a 1450 °C, 1475 °C, 1500 °C and 1525 °C silicon–molybdenum resistance furnace and subjected to sintering for 120 min, then cooled with deionized water to room temperature immediately after removal from the furnace. In order to study the effects of sintering time on cement clinker, prepared samples were firstly preheated for 30 min in a 500 °C electric muffle furnace, then rapidly transferred to a 1500 °C silicon molybdenum resistance furnace sintered for 60 min, 120 min, 180 min, and 240 min and then cooled immediately with deionized water to room temperature. After cooling, all samples were put inside an electric blast oven at 60 °C and dried until weighed.

### 2.2. Analysis and Characterization

#### 2.2.1. Free Lime Test

Properties of cement clinker can be negatively influenced by factors such as a small amount of CaO which can’t combine with acidic oxides such as SiO_2_, Al_2_O_3_, Fe_2_O_3_, and exists in the free state, called free lime f-CaO. The content of f-CaO directly indicates the quality of clinker calcination. The existence of f-CaO affects the stability and other properties of cement to varying degrees, so it is one of the main focuses of production quality control to monitor f-CaO contents. F-CaO was measured by the ethanol glycol method according to the national standard (PRC GB/T 176-2017 Methods for chemical analysis of cement) for clinkers prepared at different calcining temperatures and sintering times.

#### 2.2.2. XRD and Rietveld Quantitative Analysis

The D8 Advance X-ray diffractometer of Brucker company (Karlsruhe, Germany) was adopted, with specifications as follows: acceleration voltage 40 kV, acceleration current 40 mA, step scan with step length of 0.02°, scan time 1s per step, and the scanning angles between 5~70°. Obtained patterns were quantitatively analyzed by using GSAS software [19] for Rietveld refinement [20,21]. The crystal structure information in the sample was taken from the crystal information file (CIF) of the inorganic crystal structure database (ICSD), as shown in Table 2. Refined parameters included zero drift, background function, scale factor, peak function, grain width, cell parameters and preferred orientation. Atomic coordinates, atomic space occupation and atomic thermal vibration of all samples were not refined. The visual fitting degree of the spectrum and the error factor (Rwp) were used to determine the accuracy of the Rietveld method.

In order to minimize the interference of the preferred orientation on the intensity of the diffraction peaks, the samples subjected to an XRD test were prepared by the scatter method. In addition, 10% α-Al_2_O_3_ was added as the internal standard sample to accurately quantify the amorphous content. Since the upper surface of samples prepared by the scatter method was not at the absolute instrument zero, the diffraction summit might have shown different degrees of deviation, but this would not affect the phase judgment and the results of Rietveld quantitative analysis.

#### 2.2.3. SEM Analysis

The Quanta 200FEG scanning electron microscope made by FEI (Hillsborough, OR, USA) was used equipped with backscatter electron detector signal in a low vacuum mode at an acceleration voltage of 20 kV. Each specimen was cured with resin, and polished with alumina-alcohol solution to make the observation surface as smooth as possible.

#### 2.2.4. TG-DSC Analysis

The STA 449 F5 Jupiter synchronous thermal analyzer made by Netzsch (Selbu, Germany) was adopted. The atmosphere used was air. The heating rate was 10 °C/min, and the test range was 30~1500 °C.

#### 2.2.5. Hydration Characteristics Analysis

The TAM Air-8 thermal activity microcalorimeter (TA Instruments, New Castle, PA, USA) was adopted to measure the heat of hydration. The test temperature was 25 °C, and the water cement ratio was 0.5. The test took 3 days. Clinker and water were mixed for 30 min (after the sensor was stabilized) to collect data.

## 3. Results and Discussion

### 3.1. Free Lime Test

Measured contents of free lime(f-CaO) are shown in Figure 1.

From Figure 1a, it was found that, with the increase of sintering temperature and the extension of sintering time, the content of f-CaO in the clinker shows a downward trend. When the sintering temperature rises from 1450 °C to 1475 °C, the f-CaO content in the clinker decreases significantly. When sintering temperature is between 1475 °C and 1500 °C, the f-CaO content in the clinker decreases to a lesser extent. When the sintering temperature is higher than 1500 °C, the f-CaO content in the clinker presents a significant decline trend. From Figure 1b, the effect of sintering time on f-CaO shows a similar trend. When sintering time is between 60~120 min, the f-CaO content in the clinker shows a significant downward trend. However, when the sintering time is between 120~180 min, the f-CaO content in the clinker shows almost no change. When the sintering progresses to 180~240 min, the f-CaO content in the clinker is reduced significantly again. This is consistent with Huxing Chen’s conclusion [28].

The variation of f-CaO content (as shown in Figure 1) with sintering temperature and sintering time can be explained by the principle of solid phase reaction (see 3.3 SEM analysis for details), and can also be indirectly associated with the C_2_S content as explained in 3.2.2.

### 3.2. Crystal Types Analysis

#### 3.2.1. Phase Analysis by XRD

XRD patterns of clinkers subjected to different sintering temperatures and sintering times are shown in Figure 2.

From Figure 2, it was noted that, with the change of sintering temperature and sintering time, the peak shape of C_3_S corresponding to the window region as well as the crystal shape have changed significantly. Since the intensity of diffraction peak cannot represent phase content, phase content is not discussed in phase analysis. In addition, the diffraction peak intensity of C_3_A and Al_2_O_3_ shown a negative correlation. Since 10% Al_2_O_3_ was added as the internal standard sample, alumina reflection was very strong.

According to the literature reports, crystal shapes of C_3_S can be determined by the diffraction peak characteristics between 32°~33° and 51°~52° [7]. Figure 3 and Figure 4 show XRD patterns of clinkers subjected to different sintering temperatures (sintering time was 120 min) and sintering times (sintering temperature were 1500 °C).

Figure 3 shows C_3_S crystal symmetry changes with sintering temperatures. When the sintering temperature is between 1450 °C and 1500 °C, the diffraction peak splits into three peaks at 32°~33° and presents a shoulder peak on the left side of the diffraction peak at 51°~52°, indicating that the C_3_S crystal types are T3 and M1. When the sintering temperature is 1525 °C, the diffraction peak at 32°~33° splits into two diffraction peaks, among which there is a small shoulder peak on the each side of the right diffraction peak, and a shoulder peak on the left side of the diffraction peak at 51°~52°, indicating that the main crystal types of C_3_S are M1 and M3.

Figure 4 discloses that C_3_S crystal symmetry changes with sintering time. When the sintering time is 60~120 min, the diffraction peak splits into three at 32°~33° and presents a shoulder peak on the left side of 51°~52°, indicating that the C_3_S crystal types are T3 and M1. When the sintering time is 180 min, the diffraction peak at 32°~33° splits into two diffraction peaks, among which there is a small shoulder peak on each side of the right diffraction peak, and a shoulder peak on the left side of the diffraction peak at 51°~52°, indicating that the main crystal types of C_3_S are M1 and M3. When the sintering time is 240 min, the diffraction peak at 32°~33° divides into two diffraction peaks, among which there is a small shoulder peak on the left side of the right diffraction peak, and there is only one diffraction peak at 51°~52° without a shoulder peak, indicating that the C_3_S crystal types are M3 and R.

The results of Rietveld quantitative analysis below also support this conclusion.

#### 3.2.2. Rietveld Quantitative Analysis

The XRD patterns of seven samples subjected to different calcination conditions were refined by using GSAS software for Rietveld quantitative analysis. Taking a sample sintered at 1450 °C as an example, the result of Rietveld quantitative analysis is shown in Figure 5. The Rwp results of the seven samples are shown in Table 3, and quantitative analysis results are shown in Figure 5. The Rietveld quantitative analysis (Rwp) of the seven XRD patterns is lower than 12%, which indicates that the refining results are very reliable and support the judgment of crystal type as discussed in 3.2.1.

As shown in Figure 6, with the extension of sintering time and the increase of sintering temperature, the crystal types of C_3_S have obviously transformed, and the transformation trend was as follows: T3→M1→M3→R. At the same time, the content of C_2_S and C_3_A shown a decreasing trend, while the amorphous content shown a rising trend.

It is worth noting that the content of C_3_S and C_3_A should be 72.84% and 10.28%, according to the calculation by the Bogue method, but the fitted value is slightly lower than the calculated value. On one hand, the C_3_S formation was inadequate, but the more important reason was that the lack of sensitivity of XRD analysis forwards uncrystallized or microcrystalline phases in the clinker, which could not be crystallized due to the rapid cooling and hence could not be detected by XRD. Therefore, the phase content obtained by the Rietveld method is different from that obtained by the Bogue inverse method. Meanwhile, C_2_S was not sufficiently crystallized, and its own solid solution tendency to form in the mesophase was lower than that of C_3_S and C_3_A.This explains why the content of C_3_S is lower and the content of C_2_S was higher in Rietveld quantitative analysis [29]. There are specific images to support this conclusion in the analysis of SEM results.

### 3.3. SEM Morphology Analysis

SEM was performed using a backscattered electron method. SEM images showing changes associated with sintering temperature (sintering time was 120 min) and sintering time (sintering temperature was 1500 °C) variations are shown in Figure 7 and Figure 8.

From Figure 7, it was deduced that, with the increase of sintering temperature, the grain size presents an obvious growth trend, and the grain edge is becoming gradually flat. When the sintering temperature rises from 1450 °C to 1475 °C, the amount of liquid phase increases and the diffusion rate is faster, so the reaction rate is faster. Therefore, compared with 1450 °C, the C_3_S grains are more compact exhibiting more even grain edges. When the sintering temperature is between 1475 °C and 1500 °C, C_3_S grain boundary fusion mainly occurs at this stage, where large grains have not yet swallowed small grains, and the grain boundary has not expanded. At this time, sintering temperature is no longer the controlling factor of the solid state reaction rate, so the reaction rate does not significantly improve, and the grain morphology does not significantly change. When the sintering temperature is between 1475 °C and 1500 °C, the grain size of C_3_S increases significantly, and the high temperature is conducive to the nucleation and growth of the grain. For the same nucleation and growth rate, the high temperature requires lower saturation [30,31].

As can be seen from Figure 8, when the sintering time is 60 min, the grain size of C_3_S is very small. When the sintering time is 120 min, the grain boundary of C_3_S grain gradually contacts, the edge is flat, and the grain grows. When the sintering time is 180 min, the edge of C_3_S grain continues to expand, while the larger grains merge with the smaller grains and grow. When the sintering time is 240 min, the C_3_S grains get in contact with each other once again, the grain edges flatten, and the grain size increases again. This phenomenon revealed the law behind C_3_S grain formation and growth. Ranging from 60 to 120 min, during this period, a large number of nucleated grains are small in size and large in number. Due to the large spacing between grains, the reactions would not restrict each other. Ranging from 120 to 180 min, the boundaries of grown grains have in contact with each other at first, and the expansion of grain boundaries is blocked. At this time, the formation rate of C_3_S decreases significantly, grain boundaries are gradually fused, and large grains merge with small grains, which is also consistent with the characteristics of solid phase reaction according to the Ginsterlinger equation. From 180 to 240 min, with the further growth of the mutually annexed grains, the C_3_S formation rate is accelerated again until the larger grains come into contact again.

The SEM analysis results above show a correlation with the above f-CaO test results, which also confirm the variation trend in C_2_S content through the Rietveld quantitative analysis.

Figure 9 shows that there are many tiny grains in the mesophase between large size C_3_S grains, and the grain is too small to contribute to the diffraction peak intensity in the XRD diffraction pattern. This explains why the content of C_3_S and C_4_A in the quantitative analysis of Rietveld refinement is significantly lower than that calculated by the Bogue method.

### 3.4. TG-DSC Analysis

The raw materials shown in Table 1 were tested by DSC-TG, with heating rate 10 °C/min and temperature up to 1500 °C. The experimental results are shown in Figure 10.

Figure 10 discloses an obvious endothermal peak which appeared at 818 °C, and the mass also decreased significantly, which is associated with decomposition of CaCO_3_. The exothermic peak which appeared at 1138 °C is associated with C_2_S and C_3_A, the endothermic peak which appeared at 1340 °C is linked to the liquid phase, and the exothermic peak which appeared at 1359 °C is associated with C_3_S. It can be seen that, due to the high silicon ratio, during the formation of high alite white Portland cement clinker, the liquid phase appears at a significantly higher temperature than that of ordinary Portland cement clinker. C_3_S can only be generated in large quantities after the liquid phase appears above 1340 °C.

### 3.5. Hydration Characteristics Analysis

In terms of the rate of heat release from hydration, hydration is a matter of mass transfer kinetics. In terms of energy change of the total heat of hydration, hydration is a matter of thermodynamics. The heat flow rate curves of hydration heat for clinkers prepared at different sintering temperatures (sintering time was 120 min) and at different sintering times (sintering temperature was 1500 °C) are shown in Figure 11. Total heat release curves of the heat of hydration for clinkers prepared at different sintering temperatures (sintering time was 120 min) and at different sintering times (sintering temperature was 1500 °C) are shown in Figure 12.

From Figure 11, it can be seen that, with the increase of sintering temperature (1450~1525 °C) and the extension of sintering time (60~120 min), the accelerating period of hydration starts later. According to the theory of induction period [31], the surface activation point of C_3_S has been hydrated and formed a silicon-rich layer and a dual-electric layer. The reason for the beginning of the acceleration period is that the precipitation of CH will reduce the concentration of Ca^2+^ in the solution, which will further promote the dissolution of clinker minerals and increase the concentration of SiO_3_^2−^ in the solution. The appearance of SiO_3_^2−^ will help C-S-H to grow up. At the same time, the continuous formation and growth of C-S-H crystal nucleus will further reduce the supersaturation required for C-S-H precipitation, and the interaction between the two will lead to the accelerated hydration reaction [32,33,34]. In the acceleration period, C_3_S is dissolved in great quantities, Ca(OH)_2_ in the solution is saturated, ζ potential drops rapidly, and the hydration speeds up. During the decline period, due to the increase of hydration products, the growth space in cement slurry decreases, the hydration products intersperse and adsorb each other, the crystal growth space is significantly compressed, and the number of nucleation sites decreases. According to the above and quantitative analysis results of Rietveld, the hydration activities of different C_3_S crystals in this experiment are ranked from high to low as follows: T3→M1→M3→R.

From Figure 12, it was noted that, with the increase of sintering temperature (1450~1525 °C), the total heat release presents an increasing trend, but for the sintering time (60~240 min), the total heat release is ranked from high to low as follows: 60 min→120 min→240 min→180 min. According to Savas, Kaya [35], lattice energy can be assessed indirectly using experimental values of heat of formation, heat of ascension, heat of dissociation, ionization energy, electron affinity potential, etc. According to the above results of Rietveld quantitative analysis, we believe that the lattice energy of M- series crystals is significantly higher than that of T- series and R- series crystals.

When clinker was sintered at 1500 °C, with the increase of sintering time, the stability of C_3_S crystal structure changed first from high to low and then to high. This also explains why the pure C_3_S crystals prepared in the laboratory are all T-series crystals [36].

## 4. Conclusions

Sintering temperature and sintering time will significantly change the C_3_S crystal types in white Portland cement clinker. With the increase of sintering temperature (from 1450 to 1525 °C) and sintering time (from 60 to 240 min), f-CaO in clinker will decrease, C_3_S grain size will increase, and C_3_S crystal types will change from T3 to M types and R types which show higher symmetry.Rietveld quantitative analysis combined with an internal standard method can be used to accurately quantify the mineral composition and content of cement clinker. The differences between this method and traditional Bogue method lie in the existence of an amorphous phase.The hydration activity of C_3_S crystal ranged from high to low as follows: T3→M1→M3→R, and the lattice energy of M-series crystal was found to be higher than that of T-series and R-series crystals.As for the relationship between hydration heat and lattice energy, only a qualitative description was made here. Specific analysis can be combined with simulation analysis. Authors believe that, with the extension of sintering time, the C_3_S crystals will eventually change to T-series crystals. Compared with the original T-series crystal, these shows poorer activity and fewer defects.

## Figures and Tables

**Figure 1 materials-13-00494-f001:**
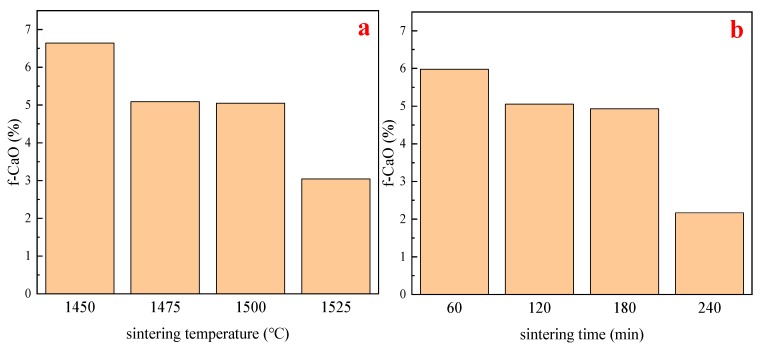
The content of f-CaO as a function of sintering conditions. (**a**) Changes in f-CaO content as a function of different sintering temperatures. (**b**) Changes in f-CaO content as a function of different sintering times.

**Figure 2 materials-13-00494-f002:**
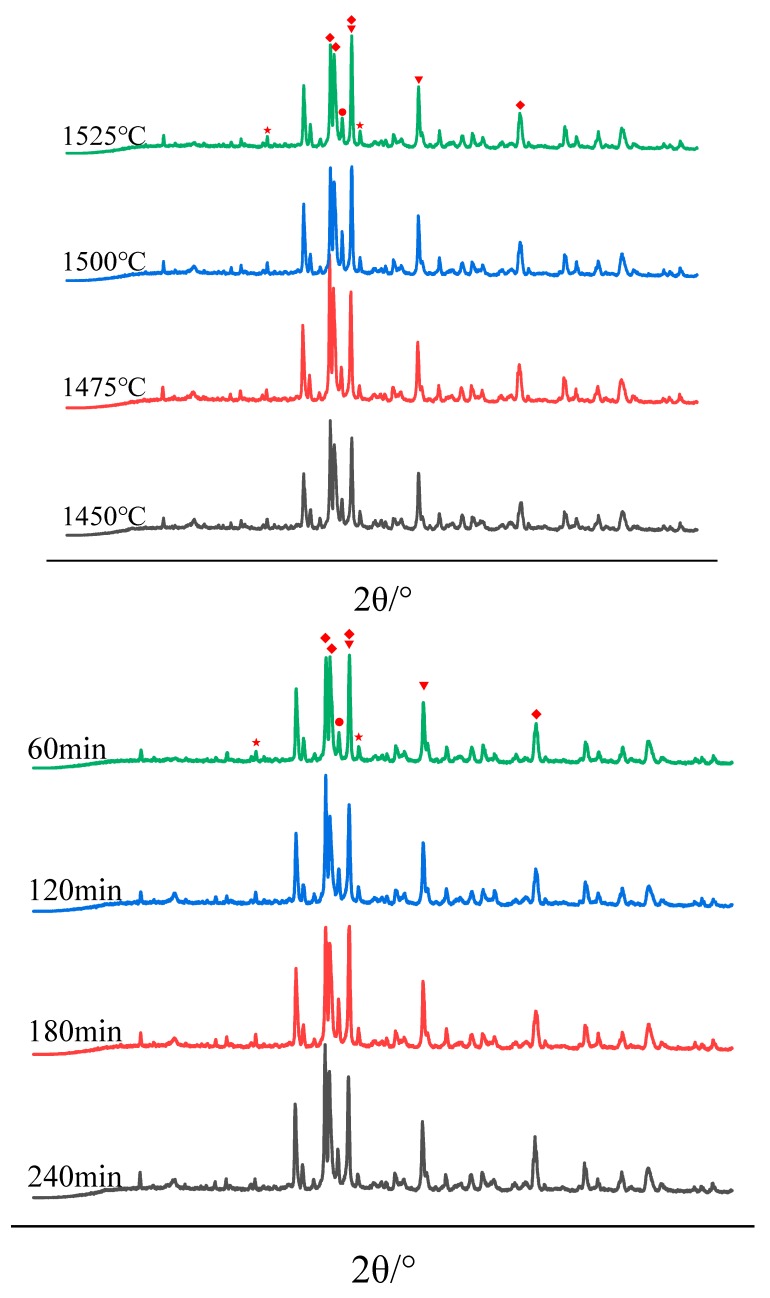
XRD patterns of clinkers subjected to different sintering temperatures and sintering times. (“◆” represents C_3_S, “▼” represents C_2_S, “●” represents C_3_A, “★” represents Al_2_O_3_).

**Figure 3 materials-13-00494-f003:**
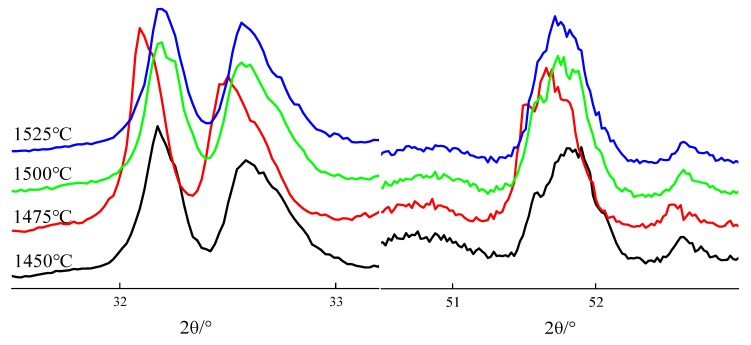
Changes in the C_3_S polymorphs as a function of different sintering temperatures.

**Figure 4 materials-13-00494-f004:**
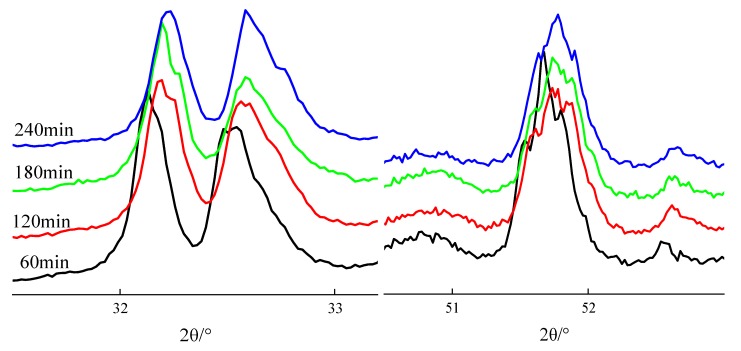
Changes in the C_3_S polymorphs as a function of different sintering times.

**Figure 5 materials-13-00494-f005:**
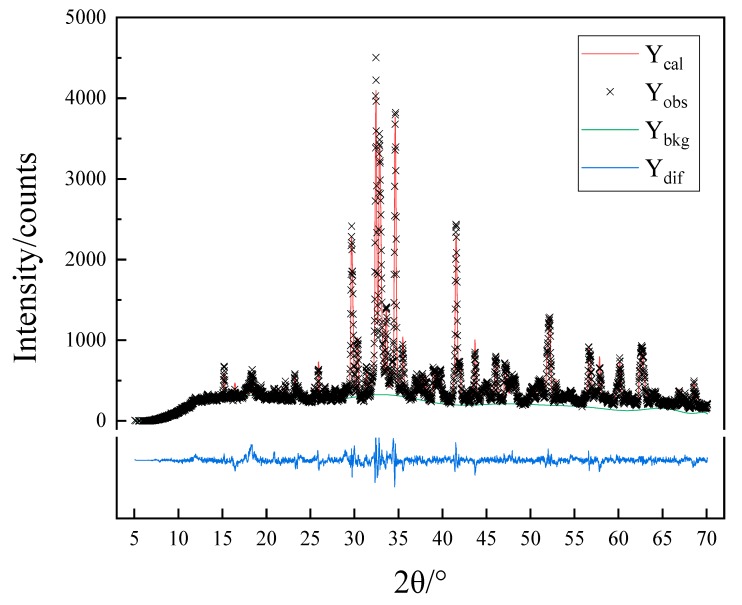
The Rietveld method applied for a sample sintered of 1450 °C.

**Figure 6 materials-13-00494-f006:**
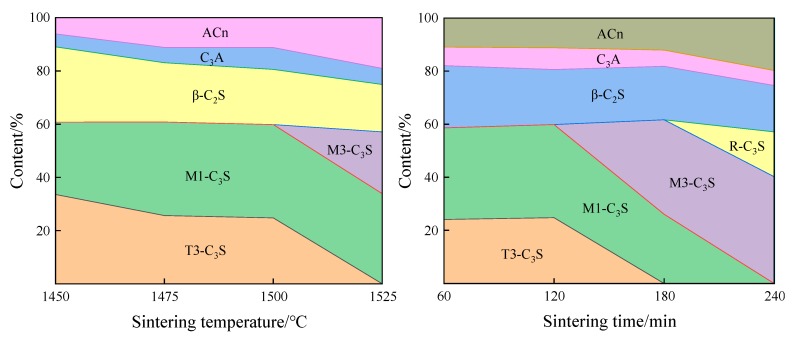
Results of the Rietveld method as a function of different sintering temperatures and times.

**Figure 7 materials-13-00494-f007:**
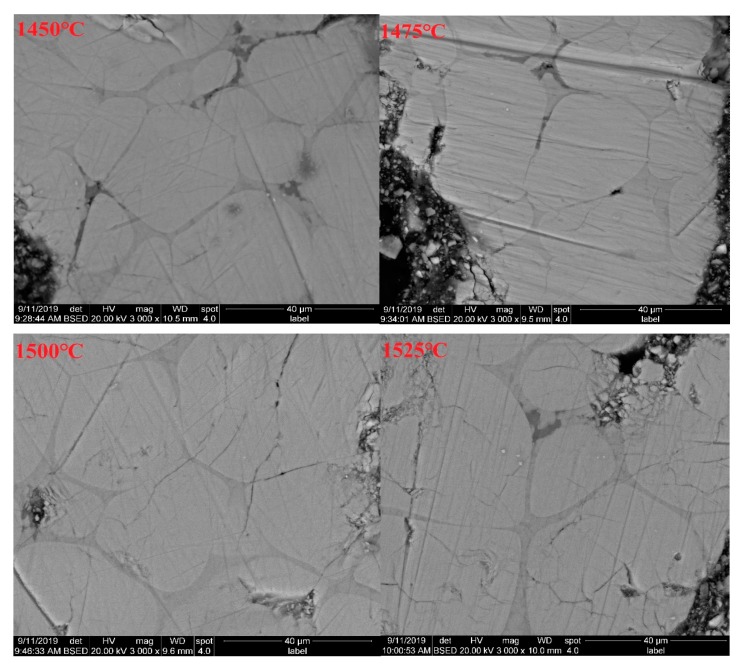
SEM of the white cement clinker subjected to different sintering temperatures.

**Figure 8 materials-13-00494-f008:**
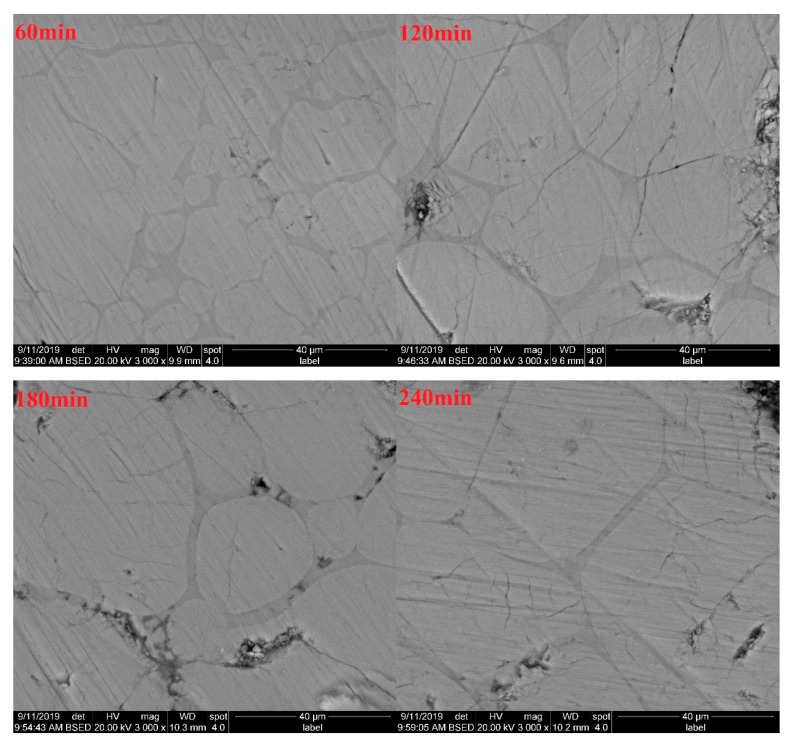
SEM of the white cement clinker subjected to different sintering time.

**Figure 9 materials-13-00494-f009:**
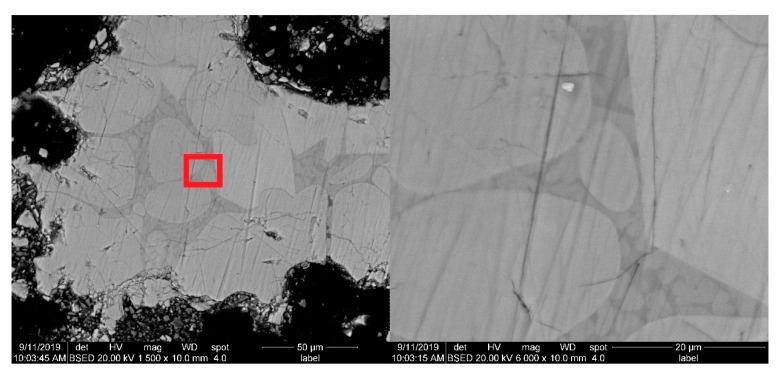
SEM image of the clearance between the C_3_S.

**Figure 10 materials-13-00494-f010:**
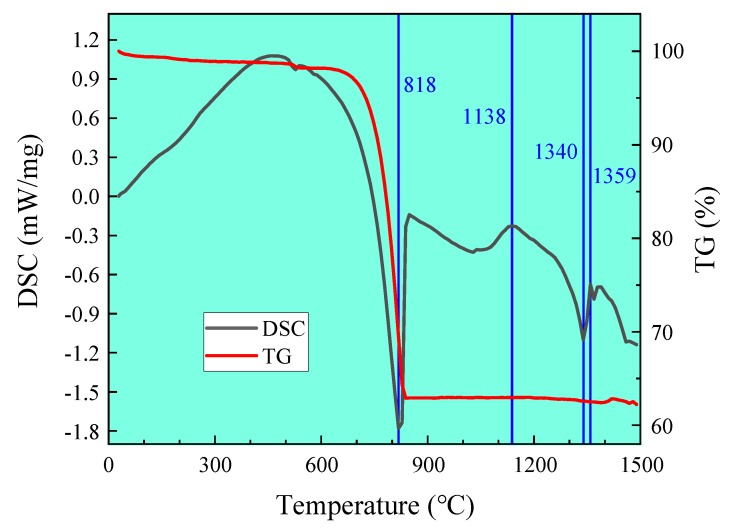
DSC-TG of cement of the white cement clinker.

**Figure 11 materials-13-00494-f011:**
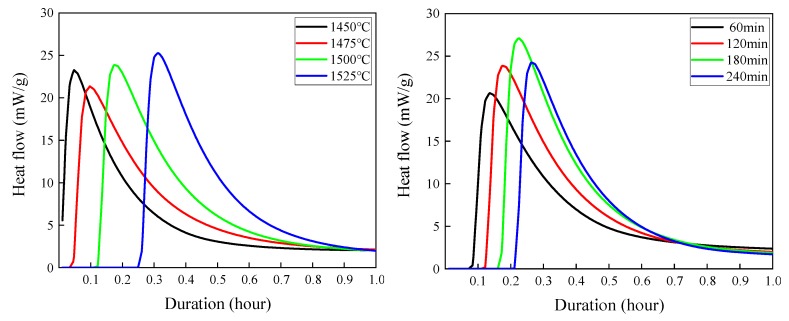
Heat flow for white cement clinker sintered at different sintering temperatures and sintering times.

**Figure 12 materials-13-00494-f012:**
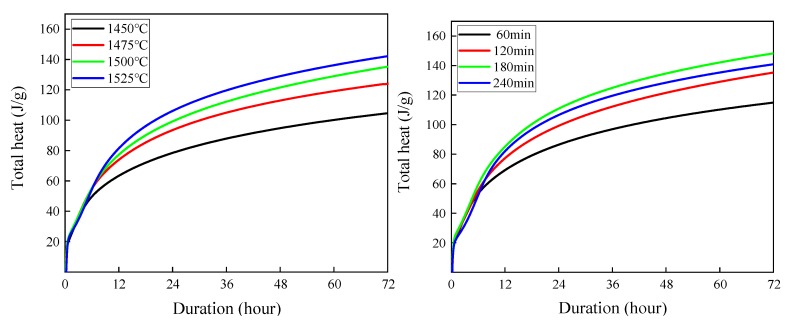
Total heat release for white cement clinker sintered at different sintering temperatures and sintering times.

**Table 1 materials-13-00494-t001:** Details for Bogue’s method of design, the chemical composition and mineral composition.

Chemical Composition		Mineral Composition	
Oxide	Wt (%)	Mineralogical constituent	Wt (%)
CaCO_3_	81.41	C_3_S	72.84
SiO_2_	16.11	C_2_S	17.35
Al_2_O_3_	2.48	C_3_A	10.28

**Table 2 materials-13-00494-t002:** Crystal structure information.

Name	Formula	Crystal System	ICSD Code	References
Alite	Ca_3_SiO_5_	Triclinic/T3	162744	Ángeles et al. [15]
		Monoclinic/M1	81100	Mumme [16]
		Monoclinic/M3	64759	De La Torre et al. [22]
		Rhombohedral/R	22501	Il’inets et al. [23]
Belite	Ca_2_SiO_4_	Monoclinic	81096	Mumme et al. [24]
Aluminate	Ca_3_Al_2_O_6_	Cubic	1841	Mondal P et al. [25]
Lime	CaO	Cubic	52783	Smith D K et al. [26]
Aluminium oxide	Al_2_O_3_	Trigonal	75559	Sawada [27]

**Table 3 materials-13-00494-t003:** The Rwp of the Rietveld method.

	60 min	120 min	180 min	240 min
1450 °C		8.88%		
1475 °C		7.66%		
1500 °C	10.61%	9.82%	11.24%	10.28%
1525 °C		9.89%

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
