# Peer review of "Effects of Calcination Conditions on the Formation and Hydration Performance of High-Alite White Portland Cement Clinker"

_materials, 2020, doi:10.3390/ma13030494_

Round 1

Reviewer 1 Report

Introduction requires scientific development.
In 53 lines at 1375 - there should be 1475.
From 84 lines, all text needs to be formatted according to MDPI guidelines.
Line 87 polished - 2x.
Fig. 1 standardize the mark spacing on the horizontal axes.
Point. 3.2 - description of the test method should be moved to point 2.
Fig. 6-8 - Scratches indicate improper polishing of test specimens.
Line 241 (OH)2.
Point 3 is a description of the test results. No scientific discussion.

Reviewer 2 Report

The subject of this manuscript is of interest for researchers in the cement and concrete field. However, several experiments need more details, and the manuscript has to be edited for English grammar and style.

Page 1 Line 10: What do you mean by "etc.", I did not see additional experiments?

P1L22: Capital letter for "Compared" not needed. Should check the whole manuscript for this similar error.

P1L31: You mention that the high-alite Portland cement "almost does not contain iron", and then you mention that "there is no C4AF in the process of clinker formation". If there is iron, where is it in your samples?

P2L53: Correct the temperature 1375C with 1475C.

P2L58: You mention that the clinkers were quenched in water, did they hydrate during this process?

P2L69: Please, reference the method used for the determination of free lime, ASTM standard, or another standard?

P2L74: what is an "atlas"?

P3L85: Change "EFI" with "FEI".

P3L87: What kind of "solution" did you use to polish the samples" Non-aqueous?

P3L92: Replace "TA/TAM" with TA Instruments TAM Air.

P3L94: You mention that "the clinker and water were mixed for 30 min (after the sensor was stabilized) to collect data", does it mean you did not record the first 30 min of the hydration process? We usually mix the clinker and water for 30sec to 1min.

P3: The font of the manuscript is different starting on line 84.

P4L120: The XRD results should show the whole XRD patterns, so the peaks of tricalcium aluminate and belite could be seen, and the purity of the samples can be discussed.

P5-Table3: Rietveld parameter Rwp should be a value between 1 and 10 if this is a "good" analysis. Your values should be reported as %, such as 0.0888 should be 8.88. Rwp below 10 is considered good, but you have Rwp around 11 and 12, a clinker phase might be missing during your interpretation.

P6L162: Bogue method does not account for impurities which may come from your raw materials.

P8-Figure7: The SEM images show marks from the polishing process.

Figure 10: The heat flow for the 4 samples shown in Figure 10 does not start at the same time, or is it actually delayed?

The experiments described in the manuscript are appropriate for the research described, but work on the interpretation is necessary, such as the full XRD patterns should be shown to detect the presence of possible additional phases (which would explain a high Rwp), or the heat flow graphs for the samples should all start at the same time, so they can be actually compared,...

Reviewer 3 Report

Reviewer report

COMMENTS

Line 30: references should be moved to line 29.

Line 41: Experimental better than experiment.

Lines 46-49: Very big characters.

Lines 84-272: Very small characters.

Lines 84-272: The format is missing. “Figure” should be added instead of Fig.

Lines 100-110: Sintering temperature effects in the clinker are well-known. Add a reference and compare your results with the literature.

Fig.2 and Fig.3 which are the XRD patterns of different sintering temperatures should be clearer. For instance, do not overlap the lines.

Lines 120&146: After 3.2.1 is written 3.3.2.

Line 155: Fig.4: Add the name of the mineralogical compounds in the graph.

Please, use subscript in the references 22, 25, 32, 35 and so on.

Please, use abbreviations in the magazine´s name. For instance, in ref. [22], it must be Cem. Concr. Res.

The paper must follow the formatting instructions. For instance, references and Tables must be rewritten as follows:

Author 1, A.B.; Author 2, C.D. Title of the article. Abbreviated Journal Name Year, Volume, page range.

Round 2

Reviewer 1 Report

The corrections have been accepted.

Author Response

thank you!

Reviewer 3 Report

Accept in present form.

Author Response

thank you!